# Erosion Resistance of Valve Core Surface Combined with WC-10Co-4Cr Coating Process under Different Pretreatments

**DOI:** 10.3390/ma15228140

**Published:** 2022-11-17

**Authors:** Lin Zhong, Zhichao Li, Guorong Wang, Haiyang He, Gang Wei, Sijia Zheng, Guihong Feng, Nana Xie, Rongyao Zhang

**Affiliations:** 1School of Mechanical Engineering, Southwest Petroleum University, Chengdu 610500, China; 2Energy Equipment Institute, Southwest Petroleum University, Chengdu 610500, China; 3Sichuan Province Science and Technology Resource Sharing Service Platform for Petroleum and Natural Gas Equipment Technology, Chengdu 610500, China; 4Nanchong Gleneng Natural Gas Co., Ltd., Nanchong 637000, China; 5Gathering and Transmission Engineering Technology Research Institute of Southwest Oil and Gas Field Company, Chengdu 610500, China; 6Pipeline Maintenance Branch of Chongqing Gas Group Co., Ltd, Chongqing 400000, China; 7Qinghai Oilfield Oil Production Plant No. 4 Production Command Center, Mangya 817500, China

**Keywords:** 3Cr13 valve core, laser ablation pretreatment, HVOF spray, WC-10Co-4Cr coating, erosion

## Abstract

The erosion of the valve core causes valve failure problems. Thus, a novel method to extend the erosion resistance of the valve was innovatively proposed, namely, nanosecond laser ablation micro-pits on the substrate surface and high velocity oxygen fuel (HVOF) spraying WC coating to extend the erosion resistance of the valve. The characterization and evaluation of the erosion resistance of the WC-sprayed coating after the pretreatment of the 3Cr13 substrate surface polishing/grit blasting/nanosecond laser ablation circular micro-dimple were conducted using the unit coupon erosion test of liquid–solid two-phase flow, followed by the test evaluation and analysis of the erosion resistance test of the WC coating after different pretreatments of the full-size valve core. Results showed that the micro-dimple pretreatment on the surface of the 3Cr13 substrate increased the contact area rate and bonding strength of the substrate and the WC coating. By taking erosion volume loss as the evaluation index, the erosion resistance of the micro-dimple pretreatment on the surface of the 3Cr13 substrate was increased by about 31.98% compared with that of the polishing pretreatment. Therefore, the new method of surface nanosecond laser texture pretreatment and HVOF-spraying WC coating can effectively improve the erosion resistance of the valve.

## 1. Introduction

Valves are an important part of the industrial pipeline conveying system, and the erosion on the valve core caused by solid particles is the key affecting factor of their service life and pipeline transmission safety. Thermal-spraying WC wear-resistant coatings combine the advantages of high hardness of WC particles and high toughness of metal bonding phases and, thus, they show good erosion resistance [1,2]. The use of these coatings is currently one of the effective ways to improve the erosion and wear resistance of valves and other equipment [3]. According to the research of Huang Da et al., the bonding etween the coating and the substrate is the key factor affecting the erosion resistance of the coating and improving the bonding between the coating and the substrate is conducive to improving the erosion resistance of the coating specimen [4,5]. Wang Huaren et al. found that the erosion resistance of HVOF-spraying WC-10Co-4Cr coating on the stainless steel substrate is better than that of stainless steel substrate [6].

The mechanical bite of the HVOF-spraying coating process is the main reason for determining the bonding properties of the substrate and the coating [7]. Therefore, the number of mechanical bite points that bond the coating to the surface of the substrate is one of the affecting factors of the bonding of the coating and the substrate. The bonding performance is better when the number of mechanical bite points and the mechanical interlocking strength formed by the coating and the substrate are greater. According to the bonding principle of mechanical bite between the substrate and the coating, the important premise for ensuring the service life of the coating specimens is the good quality of the substrate pretreatment [7,8]. Studies have shown that 80 – 90% of premature coating specimens’ failures are caused by improper substrate pretreatment [9]. Therefore, the quality of the substrate surface pretreatment process is one of the main affecting factors of the performance of HVOF spraying coatings.

Ji Chaohui et al. showed that grit blasting the substrate can effectively improve the bonding strength between the HVOF sprayed WC coating and the substrate [10]. Grit blasting is a good roughing processing method to provide bite points for the surface. However, cracks will appear on the surface of the substrate because grit blasting will cause damage to the surface of the substrate during the roughing process [11]. At the same time, the residual stress, stress concentration, and substrate fatigue will affect the mechanical properties of the coating and eventually damage the bonding performance of the substrate and the coating [12,13]. Nanosecond laser texturing of substrate surface is another effective method for surface roughening. The substrate can provide a mechanical bite point with more than the number of substrate sandblasted. At the same time, the coating is anchored in the micro-dimples on the surface of the substrate due to the existence of circular micro-dimples on the surface of the substrate; this condition further improves the contact area between the coating and the surface of the substrate, and the contact area is the main affecting factor of the bonding strength of the coating and the substrate [12,14]. Meanwhile, laser micro-processing can effectively remove surface contaminants, clean surfaces, and enhance surface activity; thus, it results in better wetting and adhesion of the substrate surface [12,15]. Improving the wetting condition of the substrate surface is an effective way to enhance the adhesion of the coating [16]. Therefore, the micro-texture pretreatment of the substrate can significantly improve the bonding strength of the coating and the substrate [12,17,18].

Thus, Bao Yumei et al. found that the pretreatment coupled bioceramic coating of circular micro-dimple on the surface of the material can effectively improve the friction and wear performance of the coating, and the dimple area ratio is the main influencing factor [19]. Qi Penghao et al. studied the anti-wear performance of the GCR15 steel disc surface texture and diamond-like carbon (DLC) coating coupling process. The results showed that the textured treatment of a substrate surface can effectively improve the friction performance of the coating, and the relatively thick DLC coatings show a better anti-wear effect [20]. Zhang Xiang et al. investigated the bonding properties of AlCrN coatings after polishing/grit blasting/texture of alloy tools, and the results showed that the textured substrate surface can increase the contact area between the substrate and the coating, and the bonding strength of the substrate and the coating is increased by about 38% compared to the grit blasting pretreatment [14]. Kedong Zhang et al. determined the effects of different pretreated surfaces on physical vapor deposition (PVD) coatings. The results showed that the regular micro/nano-texture can improve the bonding and wear performance of the substrate and the coating [7]. Substrate pretreatment is an effective method to improve the bonding performance of coatings to substrates [21,22], but its application to the erosion resistance of substrate surfaces is still rarely reported.

Thus, a new method for using HVOF to spray WC-10Co-4Cr coating with nanosecond laser ablation micro-dimple on the surface of the substrate was innovatively proposed in this study to extend the erosion resistance of the valve for solving the above-mentioned problem. First, the three-dimensional characterization of the surface polishing/grit blasting/nanosecond laser ablation circular micro-dimple pretreatment of the 3Cr13 valve core material substrate was conducted. Then, the unit erosion test coupons of the WC-10Co-4Cr coating were made by HVOF spraying, the mechanical bite performance of the coupon coating and the substrate was characterized by scanning electron microscopy (SEM), and the bonding performance of the coupon substrate and the coating under different pretreatment processes was evaluated. The liquid–solid two-phase flow erosion unit coupon test and the full-size valve erosion test were conducted by the nanosecond laser ablation micro-dimple pretreatment coupled with coating on the surface of the 3Cr13 valve core substrate. Combined with erosion mass loss, erosion volume loss, and erosion two-dimensional/three-dimensional morphology characterization, the effectiveness of nanosecond laser ablation circular micro-dimple on the surface of the valve core matrix in conjunction with HVOF-spraying WC-10CO-4Cr coating process to improve the erosion resistance of the valve was evaluated.

## 2. Experimental Design

### 2.1. Unit Erosion Test Design of WC Coating Coupon under Different Pretreatment

The substrate of the coupon sample is 3Cr13(Cr = 13.58%, C = 3.15%, Si = 0.8% and Fe = 82.47%) valve core material. First, 40 mm × 80 mm × 6 mm substrate samples were prepared. The surface hardness of the substrate was 225 HB, and the surface roughness was about Ra2.25 μm. Then, the PG-1A metallographic polishing machine was used to continuously polish the coupon samples for 2 min at a speed of 900 r/min and a force of 50 N. The grit blasting machine was used to pretreat the coupons under the conditions of 0.5 MPa pressure and 46# white corundum raw material. The fiber laser marking machine was used to conduct nanosecond laser ablation micro-processing of circular micro-dimples with a diameter of 300 μm, a depth of 40 μm, and an area ratio of about 28.52% on the surface of the coupon samples [18,23,24]. The fiber laser marking process parameters are shown in Table 1. The area bonding rate and bonding performance of the HOVF-spraying WC-10Co-4Cr coating with an average thickness of 200 μm coating after three different pretreatment processes was evaluated by the three-dimensional morphology of the substrate surface polishing/grit blasting/texture pretreatment in combination with SEM characterization. The HVOF spraying process parameters are shown in Table 2. HVOF consists of five parts: spray gun, powder feeding system, control system, water cooling system, and gas and fuel supply system. HVOF is used to mix fuel and oxygen and burn them in a specific combustion chamber or nozzle. Then, the high temperature and high speed combustion flame flow generated are used for spraying. The 1350VM WC powder produced by TAFA is composed of Co-10%, Cr-4%, and WC-86%. The powder is agglomerated and sintered, with particle size of 15–45 μm.

The schematic of the erosion test design of the liquid–solid two-phase flow of the WC-10Co-4Cr coating unit under the substrate surface polishing/grit blasting/nanosecond laser ablation pretreatment is shown in Figure 1a. The unit erosion test system is mainly composed of a fresh water tank, a plunger pump, a sand mixing tank, a sample holding fixture, and an auxiliary system that can accurately control the jet pressure, flow rate, and angle of attack. The erosion test conditions of the coating coupons are shown in Table 3. The particle size distribution of the eroded solid particles is shown in Figure 1b, and its average particle size is 131.318 μm. Under different pretreatments, the unit coupon erosion test of the WC coating samples is repeated three times. Then, the electronic balance, white light interferometer (BRUKE Comtour GT-K1, DATAPHSICS, Germany), and SEM (ZEISS EVO MA15, Germany) were used to characterize the erosion mass loss, erosion volume loss, and erosion morphology. The erosion resistance of the valve core substrate 3Cr13 surface, combined with WC-10Co-4Cr coating process under different pretreatments, was evaluated with this index.

### 2.2. Valve Erosion Test Design of Valve Core under Different Pretreatment

The full-size valve erosion test is aimed at the T40F-16 type of stop valve core. The valve core substrate is 3Cr13. Its surface hardness is 220 HB, and the surface roughness is about Ra2.23 μm. The WC-10Co-4Cr coating process of the stop valve core sealing contour surface polishing or nanosecond laser ablation with diameter of 300 μm, depth of 40 μm, dimple spacing of 200 μm, and area ratio of 28.52% circular micro-dimple pretreatment process after the HVOF spray thickness of about 200 μm is the same as the unit coupon coating samples. The design of the valve erosion test sprayed with coating after different pretreatments of the valve core is shown in Figure 2. The test conditions of the simulation test are shown in Table 4. On the basis of the mass loss before and after the erosion of the valve core and the three-dimensional morphology of Geomagic Studio measured by Optical 3D Scanner (Opticscan-plus-Q) under the same operating conditions, the erosion resistance of the spraying coating under different pretreatment processes for the top, head, and bottom of the stop valve core is measured and evaluated.

## 3. Result and Discussion

### 3.1. Three-Dimensional Morphology Characterization of Substrate Coupons under Different Pretreatments

A white light interferometer is used to scan the three-dimensional topography of different pretreated substrate surfaces, as shown in Figure 3. Figure 3a shows that the surface of the polishing pretreatment sample substrate is smooth and has no obvious defects, that is, the polishing pretreatment conducts good quality. The three-dimensional topography of the grit blasting pretreatment on the surface of the sample substrate is shown in Figure 3b. A large number of deep pits and plough ditches are observed on the surface of the sample. The surface of the sample is relatively rough, which is in line with the characteristics of the grit blasting treatment process. No residual abrasive particles are found in the deep pit. The service performance of the coating is determined by the number of residual sand particles on the surface of the grit blasting pretreatment sample. The service performance of the coating is worse when more sand particles remain [25]. That is, the quality of the pretreatment grit blasting sample is good. The three-dimensional topography of the surface of the nanosecond laser ablation circular micro-dimple sample substrate is shown in Figure 3c. The inner wall of the laser-ablated circular micro-dimple is relatively smooth, and the edge material of the micro-dimple has no obvious melting, burning, and casting layer accumulation. That is, the quality of the circular dimple microtexture sample is good. Figure 3d shows the average roughness distribution of the substrate surface under different pretreatment processes. The smallest surface roughness is the substrate polishing sample, and its average values are Rq0.664 μm and Ra0.522 μm; the largest surface roughness is the substrate texture sample, with average values of Rq9.711 μm and Ra5.131 μm; the average surface roughnesses of the grit blasting sample are Rq4.921 μm and Ra3.869 μm. The samples after surface polishing/grit blasting/texture surface treatment meet the requirements of HVOF-spraying WC-10Co-4Cr coating process.

### 3.2. Characterization of the Bonding Properties of WC-10Co-4Cr Coatings Sprayed on Coupons under Different Pretreatments

Under the same HVOF-spraying WC-10Co-4Cr coating process, the coating is densely spread on the substrate of polishing/grit blasting/texture pretreatment, as the SEM morphology of coupons shown in Figure 4. No significant porosity is observed on the surface of the coating, but a local textured morphology is found on the surface of the laser-ablated micro-dimple coating. The SEM of the cross-section of the coated samples is shown in Figure 5. The figure shows that the coating is more closely combined with the surface of the substrate, no obvious crack or gap appears, the internal structure of the coating is dense, and no obvious defect is observed. The coating thicknesses of the three different pretreatment processes are relatively uniform and completely fill the different bonding interface morphologies. The bonding plane of the samples treated by polished coating is smooth because the substrate surface is relatively smooth; the bonding plane for the samples treated by grit blasting coating is uneven and jagged due to the relatively rough surface of the substrate; the bonding interface of the samples treated by texture coating is wavy, which effectively increases the bonding area of the coating and the substrate; thus, the bonding performance of the substrate and the coating is improved [26].

### 3.3. Erosion Performance Analysis of WC-10Co-4Cr Coating Coupons under Different Pretreatments

Figure 6 shows the erosion mass loss histogram of the substrate 3Gr13-sprayed coating coupons after different pretreatments. Comparing with the blank control group (no coating) shows that the sprayed WC-10Co-4Cr coating on the substrate has excellent erosion resistance [27]. Obvious differences are observed in the erosion mass loss of HVOF-spraying coating samples after the pretreatment of the substrate 3Gr13 under polishing/grit blasting/laser ablation micro-dimple pretreatment. That is, the erosion resistance of the coating can be effectively changed by different pretreatments of the substrate. The erosion mass loss of the coating samples after the pretreatment of the circular micro-dimple of the nanosecond laser ablation design is about 11.48% higher than that of the polishing pretreated coating. This finding initially verifies that the nanosecond laser ablation of circular micro-dimples on the surface of the substrate pretreatment combined with the HVOF spraying process can improve the erosion resistance of the coating specimens.

Figure 7, Figure 8 and Figure 9 show the three-dimensional erosion morphology and SEM micrograph of the WC-10Co-4Cr coating samples after different pretreatments of the valve core 3Gr13 substrate. The figures reveal that the erosion pit is divided into two parts, and the whole is step-like, of which the step plane of the red cloud diagram is the coating erosion morphology (corresponding to point b in SEM Figure 7, Figure 8 and Figure 9), and the blue or light green depression area is the erosion morphology of the coating and the substrate bonding area (corresponding to point a in SEM Figure 7, Figure 8 and Figure 9). The average depth of erosion morphology of the coating in the polishing coating is about 196.580 μm, the average erosion morphology depth of the coating in the grit blasting coating is about 190.673 μm, and the average erosion morphology depth of the texture coating is about 183.608 μm, which is less than the thickness of the sprayed coating by 200 μm. The maximum volume loss of polishing coating sample is about 10.606 mm^3^, the volume loss of grit blasting coating sample is about 8.130 mm^3^, and the minimum volume loss of laser ablation micro-dimple coating sample is about 7.214 mm^3^. By taking erosion volume loss as the evaluation index, the erosion resistance of coating after laser texture pretreatment unit coupons is increased by about 31.98% compared with the polishing pretreatment of 3Gr13 substrate, and the erosion resistance is effectively improved by 11.27% compared with the grit blasting coating sample. This result effectively confirms that the substrate surface texture coupling coating process could effectively improve the coating specimens’ erosion resistance. Therefore, the substrate texture pretreatment has a higher bonding strength with the coating than the substrate grit blasting or polishing pretreatment. The comparison of the three-dimensional morphology of the erosion pit shows that the coating area of the polishing coating samples has the largest erosion penetration area and the substrate of the polishing coating samples suffered the most serious erosion. The coating area of the texture coating samples have the smallest erosion penetration area, and the substrate of the texture coating samples suffered the least erosion. Therefore, the substrate texture coupling coating has better erosion resistance than that sprayed before grit blasting or polishing pretreatment.

The erosion topography of Figure 7, Figure 8 and Figure 9 shows that the erosion morphology of the coating coupons under different pretreatments is circular and pit-like. A phenomenon of rushing through the coating exposed substrate occurs, but the erosion exposure area of the substrate is much smaller than the area of the erosion pit. Compared with polishing or grit blasting pretreatment, a large amount of coating residue remains in the micro-dimple texture of the erosion pit destroyed by erosion. Therefore, the substrate texture pretreatment can effectively improve the bonding performance of the substrate and the coating. The SEM micrograph of the erosion pit also shows no significant cutting marks, and particle impact pits on the surface coating are part of the erosion pit, but traces of coating peeling are observed. Meanwhile, the exposed part of the substrate of the erosion pit presents a large number of erosion pits and traces of substrate peeling. The base part is more severely eroded than the coating part.

Figure 10 shows the SEM micrograph of erosion morphology of texture coating. The erosion pit morphology of the texture coating is characterized by three steps. The uppermost step is the morphology of the unaltered part of the erosion samples, the middle step is the erosion morphology of the coupling interface of the texture coating, and the lowest layer is the erosion morphology of the sample substrate. The SEM micrograph at point B (slope of the uppermost and middle layers) shows that the erosion surface is relatively flat. The experimental erosion angle is 90°, and the angle between this surface and the jet beam is less than 90°. However, only a small number of erosion pit marks are left on this surface, and no cutting marks are left on this slope because the WC-10Co-4Cr coating is a wear-resistant coating, and the hardness is high when the sand particles impact this surface. The C points in Figure 10 show that the two materials are coupled to each other and embedded in the middle layer. The amount of material spalling in the substrate part is greater than that of the coating material when the particles in the sand-carried fluid are impacted due to the difference in erosion resistance of the two materials. Therefore, the erosion morphology SEM micrograph mainly shows “cleft lip” marks, which are relatively rough. The lowest layer (point D in the figure) is the erosion area of the substrate material. The erosion morphology SEM micrograph shows that the cracks in the erosion area are deep, and crack expansion and penetration occur, which will easily form fragments and peel off.

The surface of the texture coating samples is dimple bionic, which will form a “water cushion” in the dimple to cushion the impact of sand particles and cause the impact angle of sand particles to impact the wall surface of the micro-dimple less than 90°. The reflection angle changes due to the change in the incidence angle of some sand grains; this phenomenon will disturb the flow field, which will further reduce the impact kinetic energy of sand particles and increase the probability of collision between sand grains [21,28]. According to the theory of erosion of brittle materials, the mass loss of erosion is the largest at a 90° angle, and the particle impact velocity is one of the main affecting factors of the erosion rate [29]. Therefore, the micro-dimple morphology on the surface of the sample effectively improves the erosion resistance of the coating [21].

The erosion performance of the texture coating samples at a 90° angle of attack is shown in Figure 11. In Stage 1, the high-speed fluid carries sand particles that hit the surface of the texture coating sample vertically. Under the continuous impact of high-speed sand particles, a constantly squeezed normal force is generated on the surface of the coating sample, which results in stress cracks around the weak points inside the coating. As the erosion time increases, cracks in the weak position continue to increase, elongation and even cracks penetrate, and a coating flake fatigue spalling forms [30,31,32,33,34,35]. Therefore, the expansion and penetration of transverse cracks are the main cause of erosion material losses from wear-resistant materials such as coatings [21], which is the initial stage and coating erosion stage of texture coating erosion. In Stage 2, an erosion pit with a gradual increase in depth is formed until the depth of the erosion pit reaches the bonding surface of the coating and the substrate with the continuous action of the erosive medium and the increase in flake fatigue spalling on the surface of the coating. The coating embedded in the micro-dimple has a strong adhesion due to the mechanical bite between the micro-dimple texture and the coating. However, the substrate part of the texture edge will be the first to peel off under the continuous erosion of sand grains, which results in the exposure of the substrate part, and the substrate part will begin to be damaged. In Stage 3, the coating part has been completely eroded. The erosion area is the base part of the sample. Compared with the coating, the substrate erosion resistance is weaker, and the erosion is more serious. Therefore, a large number of erosion pits and erosion cracks are generated on the substrate surface of the erosion site, and fatigue peeling eventually occurs [36]. In the substrate erosion stage, the erosion is the most serious, and the material loss rate is the highest.

### 3.4. Erosion Test of Texture Coating Valve Core

The valve erosion working conditions are simulated, and the morphology before and after the erosion of the valve core of different pretreatment processes is obtained, as shown in Figure 12. The erosion morphology of the uncoated valve core changes the most, and obvious erosion marks are found on the top, head, and bottom surfaces of the valve core. The erosion morphology of the polishing coating valve core is the second, and slight erosion marks are observed at the junction of the top surface and the head surface. The erosion volume loss on the bottom surface of the valve core is the most serious, but it is significantly less than that of the bottom surface of the uncoated valve core. The valve core texture coating treats minimal changes in erosion morphology. The erosion volume loss at the junction of the top surface and the head surface and the erosion volume loss of the bottom surface are significantly smaller than the that of the valve core polishing coating treatment. Moreover, the texture of the top surface and the head surface is still clearly visible, and some textures are still found on the bottom surface of the valve core. The morphological diagram after erosion reveals that the most serious part of erosion is found at the bottom surface of the valve core. The three-dimensional scanning diagram after the erosion of the valve core shows that the maximum depth of the volume loss at the bottom surface of the uncoated valve core is about 13.02 mm, the maximum depth of the volume loss at the bottom surface of polishing coating valve core is about 11.49 mm, and the maximum depth of the volume loss at the bottom surface of textured coating valve core is about 7.98 mm. The mass loss of the valve core under different pretreatments in Figure 13 indicates that the blank control group (uncoated) has the largest erosion mass loss, followed by the erosion mass loss of the valve core after polishing coating pretreatment. The erosion mass loss of the valve core afte the texture coating pretreatment is minimal, which is consistent with the erosion test law of the unit coupons. Therefore, nanosecond laser texture pretreatment of the substrate is an effective method to improve the erosion resistance of the coating specimens, and the texture coating treatment of the valve core is a feasible method to improve the erosion resistance of the valve.

## 4. Conclusions

(1)The bonding interface of the texture coating samples is wavy, and the roughness of the textured substrate is higher than that of the polishing or grit blasting treatment substrate, which effectively increases the bonding area between the coating and the substrate.(2)The erosion test data of different pretreated WC-10Co-4Cr coating coupons show that, compared with polishing or grit blasting pretreatment, the volume loss is reduced by about 31.98% and 11.27%, respectively. Therefore, the texture coupling coating process on the surface of the substrate effectively improves the erosion resistance of coating specimens.(3)The valve erosion data of the full-size texture coating valve core show that, compared with the polishing or grit blasting pretreatment valve core, the volume loss depth of the bottom surface of the texture coating pretreatment valve core is reduced by 5.04 mm and 3.6 mm, respectively. This study verifies the feasibility of the laser texture treatment coupling coating process as a way to improve the erosion resistance of coating specimens, and it provides experimental support for its application.

In this study, it is proved that it is feasible to improve the bonding between coating and substrate by coupling coating with matrix texture pretreatmented substrate so as to improve the erosion resistance of coated specimens. Subsequently, the bonding properties between coating and substrate can be further optimized through the texture shape, texture layout, texture processing parameters, texture gradient design, and composite texture design so as to further improve the erosion resistance of coated specimens.

## Figures and Tables

**Figure 1 materials-15-08140-f001:**
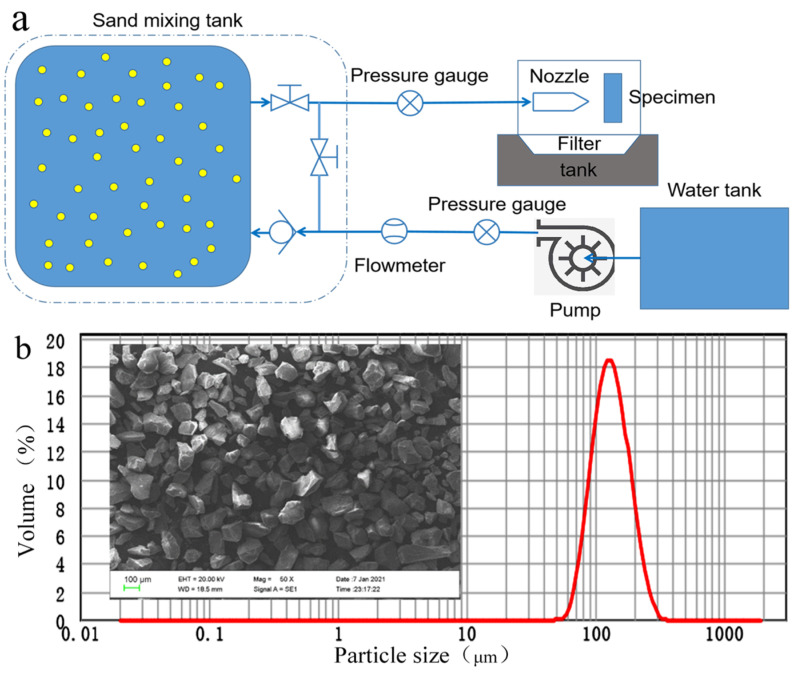
Erosion test design of liquid–solid two-phase flow of WC coating unit. (**a**) Schematic of coupon erosion test. (**b**) SEM micrograph of erosion solid particles.

**Figure 2 materials-15-08140-f002:**
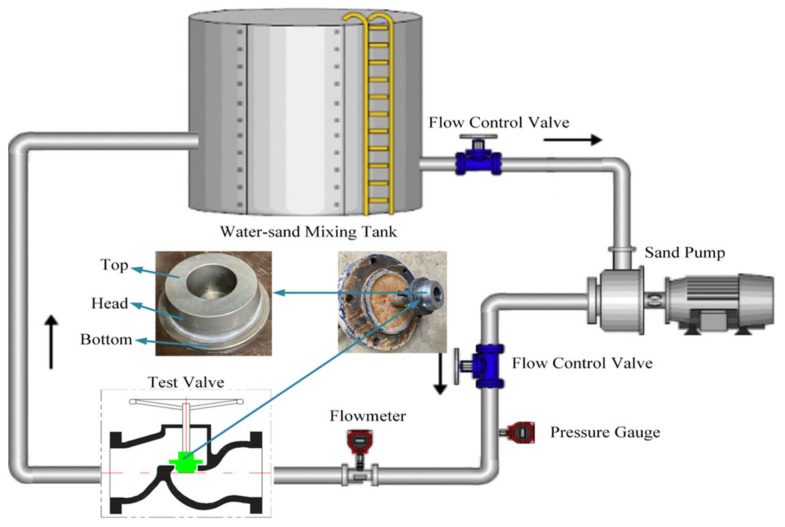
Schematic of valve erosion test design. (Black arrow indicates fluid flow direction).

**Figure 3 materials-15-08140-f003:**
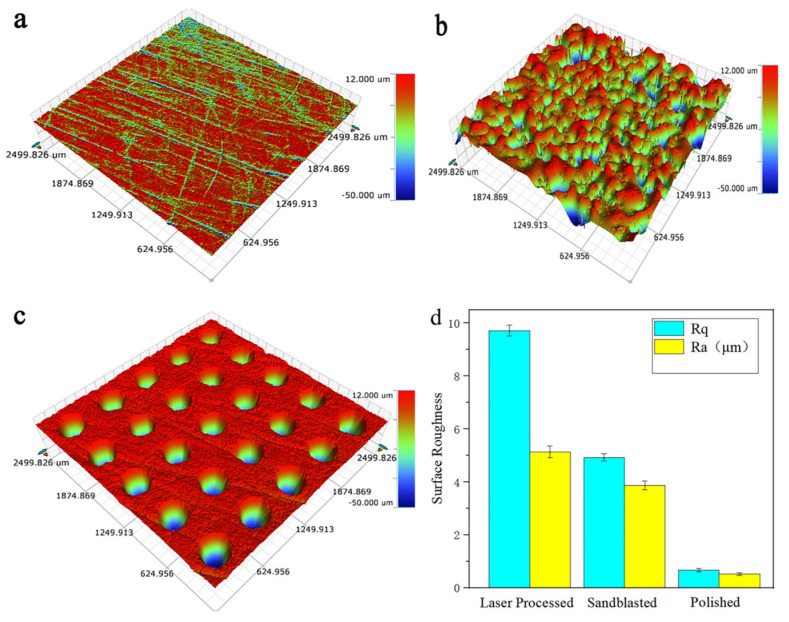
Three−dimensional morphology and roughness characterization of coupons under different pretreatments. (**a**) Three−dimensional profile of polishing. (**b**) Three−dimensional profile of grit blasting. (**c**) Three−dimensional profile of micro-dimple texture. (**d**) Comparison of roughness.

**Figure 4 materials-15-08140-f004:**
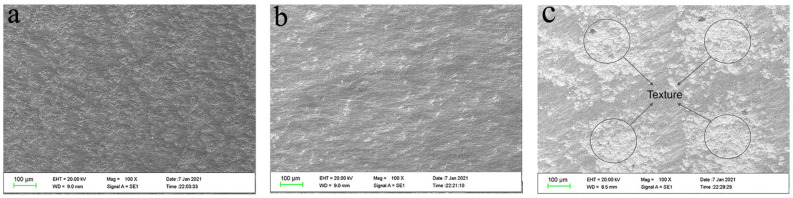
SEM of WC coating surface under different pretreatment processes. (**a**) Polished coating surface. (**b**) Sandblasted coating surface. (**c**) Textured coating surface.

**Figure 5 materials-15-08140-f005:**
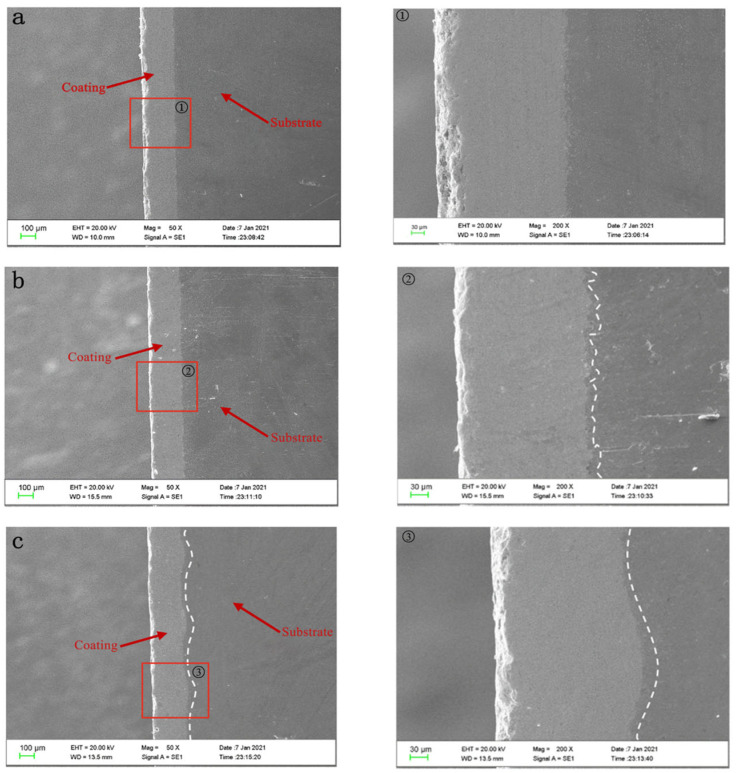
Cross-sectional SEM micrgraph of the coating after different pretreatment processes. (**a**) Coating after polishing pretreatment. (Figure ① is a partial enlarged view of the red box in Figure a) (**b**) Coating after grit blasting pretreatment. (Figure ② is a partial enlarged view of the red box in Figure b) (**c**) Coating after laser ablation texture pretreatment. (Figure ③ is a partial enlarged view of the red box in Figure c).

**Figure 6 materials-15-08140-f006:**
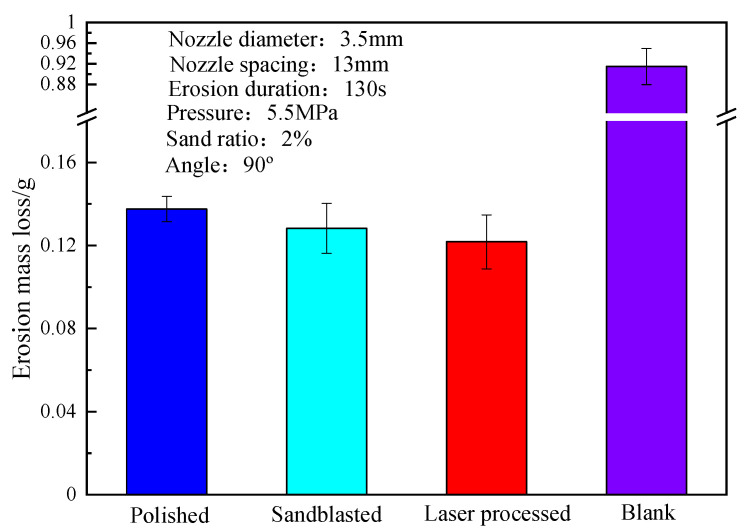
Erosion mass loss histogram of WC coating coupons under different pretreatments.

**Figure 7 materials-15-08140-f007:**
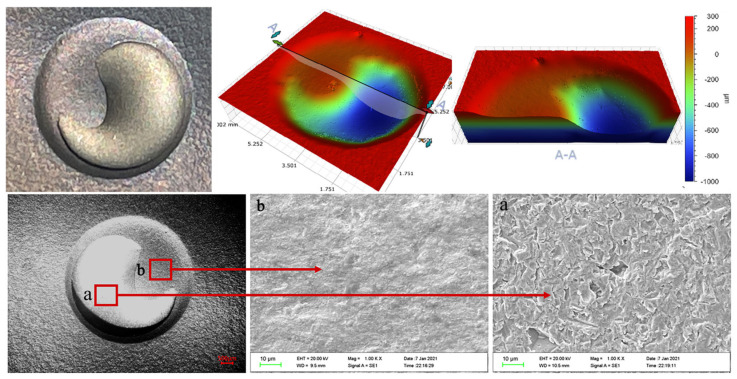
Erosion topography of polishing coating. (**a**) morphology of substrate. (**b**) morphology of coating erosion.

**Figure 8 materials-15-08140-f008:**
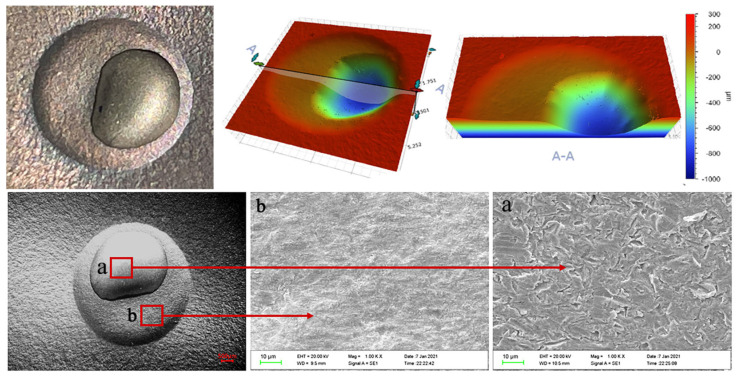
Erosion topography of grit blasting coating. (**a**) morphology of substrate. (**b**) morphology of coating erosion.

**Figure 9 materials-15-08140-f009:**
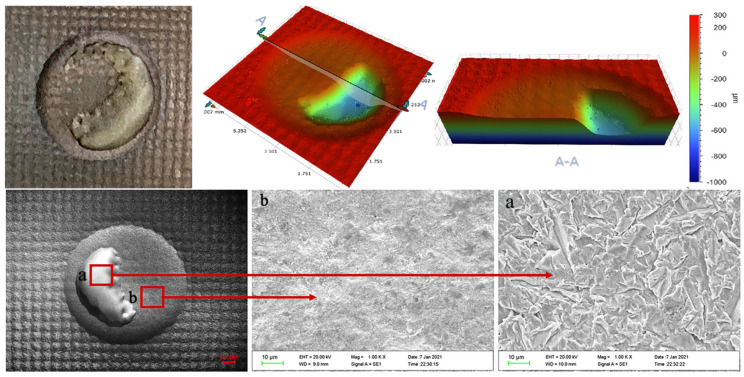
Erosion topography of micro–dimple texture coating. (**a**) morphology of substrate. (**b**) morphology of coating erosion. .

**Figure 10 materials-15-08140-f010:**
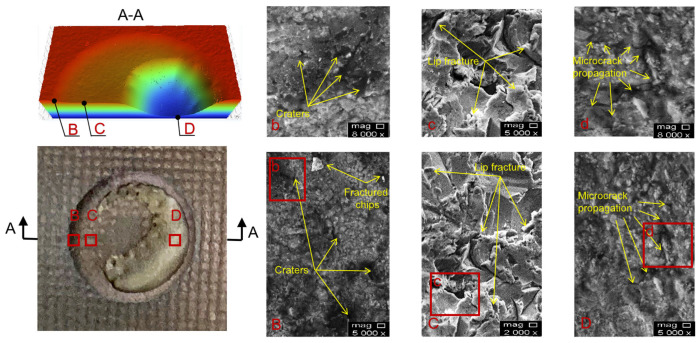
Erosion morphology of textured coating (**B**,**b** SEM micro-graph of point **B** in **A-A** diagram, **C**,**c** SEM micro-graph of point **C** in **A-A** diagram, **D**,**d** SEM micro-graph of point **D** in **A-A** diagram).

**Figure 11 materials-15-08140-f011:**
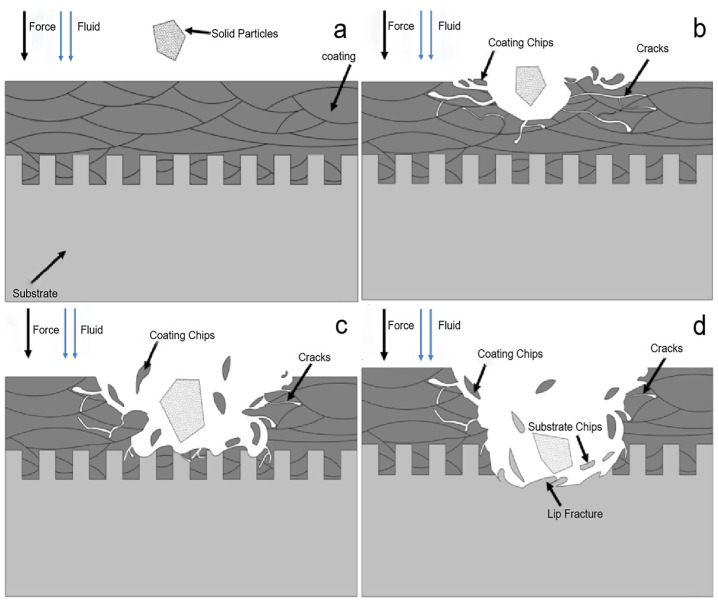
Erosion process of texture in conjunction with WC coating. (**a**) Initial state. (**b**) Coating erosion state. (**c**) Erosion state of coating-substrate bonding interface. (**d**) Substrate erosion state.

**Figure 12 materials-15-08140-f012:**
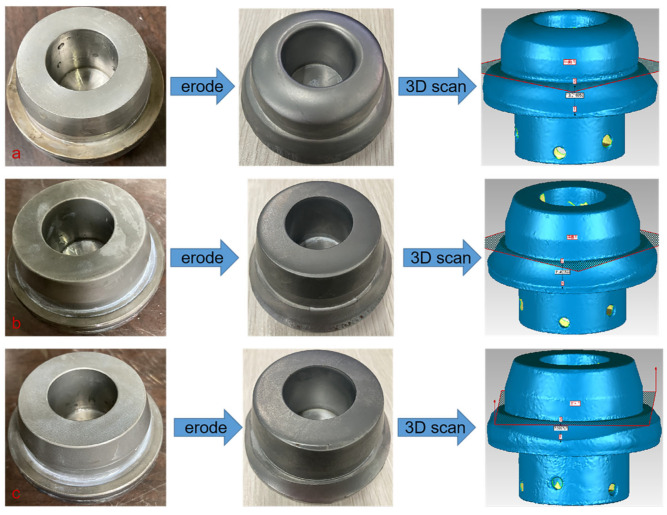
Surface morphology of valve core before and after erosion test. Group **a**: uncoated valve core. Group **b**: polish coating valve core. Group **c**: texture coating valve core.

**Figure 13 materials-15-08140-f013:**
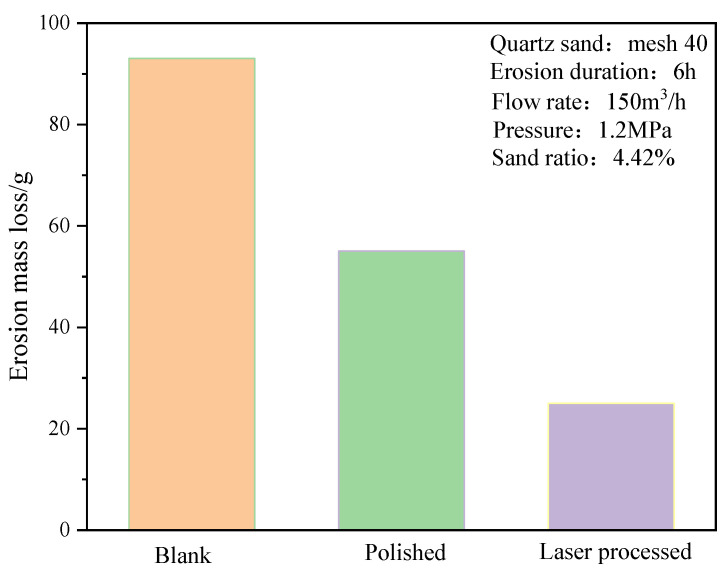
Erosion mass loss of different pretreatment valve cores.

**Table 1 materials-15-08140-t001:** Fiber laser marking Spraying parameters.

Parameter	Value
Laser	MOPA adjustable pulse width fiber laser
Wavelength(nm)	1064
Laser frequency(kHz)	800
Power(W)	10.5
Scanning speed(mm/s)	300
Processing times(times)	5

**Table 2 materials-15-08140-t002:** HVOF Spraying parameters [6].

Parameter	Value
Inlet water temperature	15.3 °C
Outlet water temperature	33 °C
Combustion chamber pressure (MPa)	0.73
Oxygen pressure (MPa)	0.99
Kerosene pressure (MPa)	0.9
Water flow (L/min)	27
Oxygen flow (m^3^/h)	36
Air pressure (MPa)	0.8
Powder delivery rate (g/min)	50
Spray distance (mm)	230
Fuel	Kerosene
powder	Tafa1350VM

**Table 3 materials-15-08140-t003:** Coupon unit erosion test working parameters.

Parameter	Value
Pressure (MPa)	5.5
Nozzle diameter (mm)	3.5
Nozzle distance (mm)	13
Volume sand ratio	2%
Erosion duration (s)	130
Erosion angle	90°

**Table 4 materials-15-08140-t004:** Valve erosion test parameters.

Parameter	Value
Pressure (MPa)	1.2
Flow rate (m^3^/h)	15
Mass sand ratio	4.42%
Erosion duration (h)	6
Quartz sand (mesh)	40
Valve opening	3 mm

## Data Availability

The data presented in this study are available on request from the corresponding author.

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
