# Peer review of "Erosion Resistance of Valve Core Surface Combined with WC-10Co-4Cr Coating Process under Different Pretreatments"

_materials, 2022, doi:10.3390/ma15228140_

Round 1
Reviewer 1 Report
In the present study, the nanosecond laser pattering is suggested as an alternative to the conventional grit blasting for the preparation of the substrate surface to improve HVOF coating adhesion and its erosion resistance. The study is interesting and is worthy to be considered for publication. However, I suggest to make some clarifications in order to further improve its value.
1. Introduction. There is a lack of consistency in the introduction. The main idea of the study, as I see, is the application of laser surface pattering to provide some developed morphology/texture to the substrate surface prior to the HVOF spraying and to improve bond strength between the protective coating and the substrate in such a way. As a result, the improve in erosion resistance is expected. For what purpose do the authors cite sources 14, 15, 16 and 17? I suggest revising your introduction in a light of the problem to be solved. Also, could authors explain clearer the relation between the coating adhesion and erosion protection (the mechanism)?
2. Line 47-48. Please, revise the sentence: Ji Chaohui et al. found that, after sandblasting the substrate, HVOF spraying can effectively improve the bonding strength of coating to the substrate. Do the authors mean that “HVOF spraying can effectively improve the bonding strength” – not sandblasting?
3. Line 54. Please, explain what is DLC.
4. Please, provide chemical composition of the substrate material and powder for spraying. The brief general characterisation of the used spraying powder would be welcome as well (particles size and morphology, powder manufacturing method).
5. The following dimensions of the dimple and laser pattering parameters are given: 300 μm in diameter, 40 μm in diameter, dimples spacing - 200 μm, area ratio – 28.52 %. Why were such parameters selected? Also, how was the parameters of grit blasting process chosen (including abrasive size)?
6. Table 1. To define prefixes, please, use symbols by SI system: „kHz“ instead of „KHz“. What does it mean „Processing times(times)“. Do the authors mean the number of applied pulses for a dimple?
7. Please, provide the description of the characterisation methods (SEM, 3D profilometry and other, if any) used in the study (including the apparatus and parameters).
8. Please, provide description for HVOF spraying equipment.
9. Line 155. Please, check subsection number and title.
10. Line 163. Sand is not used in the present study. So, I suggest to use term “abrasive particles” or “corundum particles” instead of “sand particles” and “grit blasting” instead of “sandblasting”.
11. Line 177. What exactly requirements do the authors mean?
12. Line 219. The authors stated that "cracks will appear on the surface of the substrate...". Do authors have any evidence for this statement? Or provide, please, the reference.
13. Line 230. The term “machining” or “micromachining” ir more suitable to describe mechanical processing. I suggest using term “processing”/ “micro-processing” for laser processes.
14. Lines 217-235: the text presented is more suitable for Introduction – to base why laser processing is suggested in the present study instead of conventional surface roughening by grit blasting, to base hypothesis to be checked and confirmed or refuted by study results. Here, authors have to discuss and explain results obtained in this study.
15. Line 238. Figure 6 does not contain any curves.
16. Lines 247-250. The statement that “5.07% erosion resistance improvement verifies that laser ablation technique can effectively improve the erosion resistance” is questionable. 5% improvement is no so high and is less than standard deviation given for both the grit-blasted and laser-processed specimens. I suggest avoiding so strong statements, when evaluating this result.
17. Figure 6 caption: there is no any curves in this Figure. Please, correct the caption.
18. Line 253. I suggest using term “SEM image” or “SEM micrograph” instead of “SEM diagram”.
19. Conclusions: Only one variant (dimple size, depth, and spacing) of laser pattering was applied in this work. The improvement of erosion resistance was evaluated compared with polished and grit blasted substrates. The difference between laser pattering and polishing is significant; it is absolutely expected and such comparison is not absolutely adequate, since the polishing is not typically used for surface preparation for thermal spray process. The difference between the erosion resistance of grit-blasted (typically used for substrate preparation) and laser-processed substrates is less significant and it is 5–11 %. Taking into account that the grit blasting parameters were not optimized in this study, the difference may be even smaller. Therefore, a question for authors: Can be the more significant erosion resistance improvement reached with laser pattering? If yes, how? How do the authors see the further development of this pre-treatment method? I suggest supplementing the conclusion part of the article with relevant recommendations.
Author Response
Dear Editors:
On behalf of my co-authors, we thank you very much for giving us an opportunity to revise our manuscript. We appreciate editors and reviewers very much for their positive and constructive comments and suggestions on our manuscript entitled “Erosion Resistance of Valve Core Surface Combined with WC-10Co-4Cr Coating Process under Different Pretreatments”. (Research Article, materials-2007936).
We have carefully studied the comments of the editors and reviewers and have made revisions, which are marked in red on paper. We have tried our best to revise our manuscript according to the comments. The revised version has been submitted.
We would like to express our great appreciation to you and reviewers for comments on our paper. Looking forward to hearing from you.
Thank you and best regards.
Yours sincerely,
Zhichao Li
Corresponding author:
Name: Zhong Lin
- mail: [email protected]
Dear Editors and Reviewers:
Thank you for the editors’ and reviewers’ comments concerning our manuscript entitled “Erosion Resistance of Valve Core Surface Combined with WC-10Co-4Cr Coating Process under Different Pretreatments”. (Research Article, materials-2007936).
Those comments are all valuable and very helpful for revising and improving our paper, as well as the important guiding significance to our research. We have studied comments carefully and have made corrections which we hope meet with approval. Revised portions are marked in red on paper. We also polished other parts that we didn't think were appropriate. The main corrections in the paper and the responses to the reviewer’s and editors’ comments are as follows:
Revision - authors’ response
Comment 1: Introduction. There is a lack of consistency in the introduction. The main idea of the study, as I see, is the application of laser surface pattering to provide some developed morphology/texture to the substrate surface prior to the HVOF spraying and to improve bond strength between the protective coating and the substrate in such a way. As a result, the improve in erosion resistance is expected. For what purpose do the authors cite sources 14, 15, 16 and 17? I suggest revising your introduction in a light of the problem to be solved. Also, could authors explain clearer the relation between the coating adhesion and erosion protection (the mechanism)?
Response:Thank you very much for your careful guidance and great help to the improvement of our manuscript. According to comment 14, add part of that paragraph to the introduction. In addition, the mechanism of the relationship between erosion and coating adhesion is added in the introduction, and the introduction structure is readjusted. We have made some improvements in the Introduction in the revision and marked them in red.
Comment 2: Line 47-48. Please, revise the sentence: Ji Chaohui et al. found that, after sandblasting the substrate, HVOF spraying can effectively improve the bonding strength of coating to the substrate. Do the authors mean that “HVOF spraying can effectively improve the bonding strength” – not sandblasting?
Response:Thank you for your instructive suggestions. We have revised the sentence to ‘Ji Chaohui et al. shows that grit blasting the substrate can effectively improve the bonding strength between the HVOF sprayed WC coating and the substrate’ marked them in red in the revision.
Comment 3: Line 54. Please, explain what is DLC.
Response:Thank you for your instructive suggestions. We have revised the ‘DLC’ to ‘diamond Like carbon(DLC) ’ marked them in red in the revision.
Comment 4: Please, provide chemical composition of the substrate material and powder for spraying. The brief general characterisation of the used spraying powder would be welcome as well (particles size and morphology, powder manufacturing method).
Response:Thank you for your valuable and thoughtful comments. The chemical components of the substrate 3Cr13 are Cr-13.58%, C-3% and Si-0.8%. The 1350VM WC powder produced by TAFA is composed of Co-10% , Cr-4% and WC-86%. The powder is agglomerated and sintered, with particle size of 15-45 μm . It’s marked in red in the revision.
Comment 5: The following dimensions of the dimple and laser pattering parameters are given: 300 μm in diameter, 40 μm in diameter, dimples spacing - 200 μm, area ratio – 28.52 %. Why were such parameters selected? Also, how was the parameters of grit blasting process chosen (including abrasive size)?
Response:Thank you for your valuable and thoughtful comments. The micro-dimples texture has good erosion resistance[Z.W. Han, J.Q. Zhang, C. Ge, C.F. Wang, Q.L. Ren, Gas-solid erosion wear on bionic configuration surface[J]. Journal of Jilin University (Engineering and Technology Edition)6(2009)1512-1515 doi: 10.13229/j.cnki.jdxbgxb2009.06.005) ]. The micro-dimples texture has good friction performance when the surface density is between 5% and 35% [Wang Lili, Guo Shaohui, Wei Yuliang, et al. Optimization research on the lubrication characteristics for friction pairs surface of journal bearings with micro texture[J]. Meccanica, 2019, 54(8): 1135– 1148. doi: 10.1007/s11012-019-01015-1]. The micro-dimples texture with a diameter of 300 μm, a depth of 40 μm has better coating adhesion and wear resistance. [A.X. Feng, B. Wang, Y. He, R. Yang, Y. Liu, G.Q. Zhong, Study on Influence of Laser Micro-texture on Bonding Properties of Cemented Carbide TiAIN Coating[J]. Hot Working Technology 46(2017)155-158 . doi:10.14158/j.cnki.1001-3814.2017.14.045]
Comment 6: Table 1. To define prefixes, please, use symbols by SI system: „kHz“ instead of „KHz“. What does it mean „Processing times(times)“. Do the authors mean the number of applied pulses for a dimple?
Response: Thank you for your careful reading of our manuscript. We have revised the ‘KHz’ to ‘ kHz’ marked them in red in the revision. Processing times(times) means the number of applied pulses for a dimple.
Comment 7: Please, provide the description of the characterisation methods (SEM, 3D profilometry and other, if any) used in the study (including the apparatus and parameters).
Response: Thank you for your instructive suggestions. We have added each measuring equipment model, Each micrgraph is marked with a scale. It’s marked in red in the revision.
Comment 8: Please, provide description for HVOF spraying equipment.
Response: Thank you for your instructive suggestions. HVOF consists of five parts: spray gun, powder feeding system, control system, water cooling system, gas and fuel supply system. HVOF is to mix fuel and oxygen and burn them in a specific combustion chamber or nozzle.Then, the high temperature and high speed combustion flame flow generated is used for spraying. The above sentences has been added to the paper and marked in red.
Comment 9: Line 155. Please, check subsection number and title.
Response: Thank you for your careful reading of our manuscript. We have corrected the sentence to ’3.1 3D Morphology Characterization of Substrate Coupons under Different Pretreatments’.
Comment 10: Line 163. Sand is not used in the present study. So, I suggest to use term “abrasive particles” or “corundum particles” instead of “sand particles” and “grit blasting” instead of “sandblasting”.
Response: Thank you for your careful reading of our manuscript. We have revised the ‘sand particles’ and ‘sandblasting’ to ‘ kHz’ and ‘grit blasting’ marked them in red in the revision.
Comment 11: Line 177. What exactly requirements do the authors mean?
Response: Thank you for your instructive suggestions. This requirement for the substrate surface comes from the HVOF processing plant. For example, the surface of the substrate is clean and free of oil, rust, paint, etc.
Comment 12: Line 219. The authors stated that "cracks will appear on the surface of the substrate...". Do authors have any evidence for this statement? Or provide, please, the reference.
Response: Thank you for your instructive suggestions. According to the article [11]X.Y Jiang, Y.P Wan, Herbert Herman, Sanjay Sampath, Role of condensates and adsorbates on substrate surface on fragmentation of impinging molten droplets during thermal spray.Thin Solid Films.385(2001)132-141.https://doi.org/10.1016/S0040-6090(01)00769-6. grit blasting will cause some cracks on the substrate. We have cited this article.
Comment 13: Line 230. The term “machining” or “micromachining” ir more suitable to describe mechanical processing. I suggest using term “processing”/ “micro-processing” for laser processes.
Response: Thank you for your careful reading of our manuscript. We have revised the ‘machining’ or ‘micromachining’ to ‘ processing’ or ‘micro-processing’ marked them in red in the revision.
Comment 14: Lines 217-235: the text presented is more suitable for Introduction – to base why laser processing is suggested in the present study instead of conventional surface roughening by grit blasting, to base hypothesis to be checked and confirmed or refuted by study results. Here, authors have to discuss and explain results obtained in this study.
Response: Thank you very much for your careful guidance and great help to the improvement of our manuscript. We have added this content to the introduction and adjusted other parts of the introduction. We have made some improvements in the Introduction in the revision and marked them in red.
Comment 15: Line 238. Figure 6 does not contain any curves
Response: Thank you for your careful reading of our manuscript. We have revised the ‘curves’ to ‘ histogram’ marked them in red in the revision.
Comment 16: Lines 247-250. The statement that “5.07% erosion resistance improvement verifies that laser ablation technique can effectively improve the erosion resistance” is questionable. 5% improvement is no so high and is less than standard deviation given for both the grit-blasted and laser-processed specimens. I suggest avoiding so strong statements, when evaluating this result
Response: Thank you for your instructive suggestions. We have revised this sentence to avoid using such a strong tone. Detailed sentence have been marked in red in the revised article.
Comment 17: Figure 6 caption: there is no any curves in this Figure. Please, correct the caption
Response: Thank you for your careful reading of our manuscript. Thank you for your careful reading of our manuscript. We have revised the ‘curves’ to ‘ histogram’ marked them in red in the revision.
Comment 18: Line 253. I suggest using term “SEM image” or “SEM micrograph” instead of “SEM diagram”.
Response: Thank you for your careful reading of our manuscript. Thank you for your careful reading of our manuscript. We have revised the ‘SEM image’ or ‘SEM micrograph’ to ‘SEM diagram’ marked them in red in the revision.
Comment 19: Conclusions: Only one variant (dimple size, depth, and spacing) of laser pattering was applied in this work. The improvement of erosion resistance was evaluated compared with polished and grit blasted substrates. The difference between laser pattering and polishing is significant; it is absolutely expected and such comparison is not absolutely adequate, since the polishing is not typically used for surface preparation for thermal spray process. The difference between the erosion resistance of grit-blasted (typically used for substrate preparation) and laser-processed substrates is less significant and it is 5–11 %. Taking into account that the grit blasting parameters were not optimized in this study, the difference may be even smaller. Therefore, a question for authors: Can be the more significant erosion resistance improvement reached with laser pattering? If yes, how? How do the authors see the further development of this pre-treatment method? I suggest supplementing the conclusion part of the article with relevant recommendations.
Response: Thank you very much for your careful guidance and great help to the improvement of our manuscript. We have adjusted the conclusion to avoid it being too large. In addition, a new paragraph is added to the conclusion, and the prospect of the substrates textured coupling coating to improve the erosion resistance of coated specimens is presented.
Once again, thank you very much for your comments and suggestions.

Reviewer 2 Report
Comments:
“Erosion Resistance of Valve Core Surface Combined with WC- 2 10Co-4C Coating Process under Different Pretreatments”
1. Please check the grammar and typo mistakes.
2. Use subscript for terms like for Cr13
3. Line 24, increased the area contact rate (correct grammar like contact area rate not area contact!)
4. Donot use “/” like in keywords section
5. Line 38, use comma in reference [1,2] (not [1.2])
6. Figure 1 b caption: The word characterization is general applied for various kind of investigations. Use the appropriate term very relevant to graph
7. Figure 10: there are 8 parts of this figure. Please make them as a,b,c,d,….. and h and give captions to each part
8. Please reduce the conclusions, it is too large.
9. References: Most of the references are older than 2018, please add references from recent years.
Author Response
Dear Editors:
On behalf of my co-authors, we thank you very much for giving us an opportunity to revise our manuscript. We appreciate editors and reviewers very much for their positive and constructive comments and suggestions on our manuscript entitled “Erosion Resistance of Valve Core Surface Combined with WC-10Co-4Cr Coating Process under Different Pretreatments”. (Research Article, materials-2007936).
We have carefully studied the comments of the editors and reviewers and have made revisions, which are marked in red on paper. We have tried our best to revise our manuscript according to the comments. The revised version has been submitted.
We would like to express our great appreciation to you and reviewers for comments on our paper. Looking forward to hearing from you.
Thank you and best regards.
Yours sincerely,
Zhichao Li
Corresponding author:
Name: Zhong Lin
- mail: [email protected]
Dear Editors and Reviewers:
Thank you for the editors’ and reviewers’ comments concerning our manuscript entitled “Erosion Resistance of Valve Core Surface Combined with WC-10Co-4Cr Coating Process under Different Pretreatments”. (Research Article, materials-2007936).
Those comments are all valuable and very helpful for revising and improving our paper, as well as the important guiding significance to our research. We have studied comments carefully and have made corrections which we hope meet with approval. Revised portions are marked in red on paper. We also polished other parts that we didn't think were appropriate. The main corrections in the paper and the responses to the reviewer’s and editors’ comments are as follows:
Revision - authors’ response
Comment 1: Please check the grammar and typo mistakes.
Response: Thank you very much for your careful guidance and great help to the improvement of our manuscript. We have corrected some sentences with grammatical errors and some sentences with unclear meaning. We also corrected some misspelled words, as well as some words that are not applicable. Those marked in red in the revision.
Comment 2: Use subscript for terms like for Cr13
Response: Thank you for your careful reading of our manuscript. Based on this article [Effect of Tempering Treatment on Atmospheric Corrosion Behavior of 3Cr13 Martensitic Stainless Steel in Marine Environment[J]], 3Cr13 is the abbreviation of martensitic stainless steel, without subscript,so as to others.
Comment 3: Line 24, increased the area contact rate (correct grammar like contact area rate not area contact!)
Response: Thank you for your valuable and thoughtful comments. We have revised the ‘the area contact rate’ to ‘the contact area rate ’ marked them in red in the revision. And, we have corrected some sentences with grammatical errors and some sentences with unclear meaning. We also corrected some misspelled words, as well as some words that are not applicable. Those marked in red in the revision.
Comment 4: Do not use “/” like in keywords section
Response: Thank you for your instructive suggestions. We have revised the ‘Laser ablation/polishing/sandblasting pretreatment ’ to ‘Laser ablation pretreatment’ marked them in red in the revision.
Comment 5: Line 38, use comma in reference [1,2] (not [1.2])
Response: Thank you for your valuable suggestions. We and modify all similar problems。Those marked in red in the revision.
Comment 6: Figure 1 b caption: The word characterization is general applied for various kind of investigations. Use the appropriate term very relevant to graph
Response: Thank you very much.e have revised the ‘b Characterization of erosion solid particles
’ to ‘ b SEM micrograph of erosion solid particles ’ marked them in red in the revision.
Comment 7: Figure 10: there are 8 parts of this figure. Please make them as a,b,c,d,….. and h and give captions to each part
Response: Thank you for your valuable and thoughtful comments. Considering that each graph has a one-to-one correspondence, we mark the those figure as B, b, C, c, D and d . We have provided a captions to each part. Those marked in red in the revision.
Comment 8: Please reduce the conclusions, it is too large.
Response:Thank you very much for your careful guidance and great help to the improvement of our manuscript. We have simplified the conclusion and added some prospects. Those marked in red in the revision.
Comment 9: References: Most of the references are older than 2018, please add references from recent years.
Response: Thank you for your valuable and thoughtful comments. We have add some references from recent years.Those marked in red in the revision.
Once again, thank you very much for your comments and suggestions.

Round 2
Reviewer 1 Report
Dear Authors.
I see that the revision of the manuscript was performed taking into consideration all my suggestions. Please, make three corrections more:
1. line 82: replace "Like" with "like".
2. line 113: specify, please, that the substrate material is steel (if so) and add iron to the composition.
3. Also, check, please, font size in tables.
I will recommend accepting your manuscript after these corrections.
Author Response
Dear Editors:
On behalf of my co-authors, we thank you very much for giving us an opportunity to revise our manuscript. We appreciate editors and reviewers very much for their positive and constructive comments and suggestions on our manuscript entitled “Erosion Resistance of Valve Core Surface Combined with WC-10Co-4Cr Coating Process under Different Pretreatments”. (Research Article, materials-2007936).
We have carefully studied the comments of the editors and reviewers and have made revisions, which are marked in red on paper. We have tried our best to revise our manuscript according to the comments. The revised version has been submitted.
We would like to express our great appreciation to you and reviewers for comments on our paper. Looking forward to hearing from you.
Thank you and best regards.
Yours sincerely,
Zhichao Li
Corresponding author:
Name: Zhong Lin
- mail: [email protected]
Dear Editors and Reviewers:
Thank you for the editors’ and reviewers’ comments concerning our manuscript entitled “Erosion Resistance of Valve Core Surface Combined with WC-10Co-4Cr Coating Process under Different Pretreatments”. (Research Article, materials-2007936).
Those comments are all valuable and very helpful for revising and improving our paper, as well as the important guiding significance to our research. We have studied comments carefully and have made corrections which we hope meet with approval. Revised portions are marked in red on paper. We also polished other parts that we didn't think were appropriate. The main corrections in the paper and the responses to the reviewer’s and editors’ comments are as follows:
Revision - authors’ response
Comment 1: line 82: replace "Like" with "like".
Response: Thank you for your careful reading of our manuscript. We have revised the ‘Like’ to ‘ like’ marked them in red in the revision.
Comment 2: line 113: specify, please, that the substrate material is steel (if so) and add iron to the composition. We have added the substrate element Fe=82.47%. It marked in red in the revision.
Response: Thank you very much for your careful guidance and great help to the improvement of our manuscript.
Comment 3: Also, check, please, font size in tables.
Response: Thank you for your valuable and thoughtful comments. We have modified font size in tables. Those marked in red in the revision.
Once again, thank you very much for your comments and suggestions.
